# The Effect of Cu, Zn, Cd, and Pb Accumulation on Biochemical Parameters (Proline, Chlorophyll) in the Water Caltrop (*Trapa natans* L.), Lake Skadar, Montenegro

**DOI:** 10.3390/plants9101287

**Published:** 2020-09-29

**Authors:** Dragana Petrovic, Sladjana Krivokapic

**Affiliations:** Department of Biology, Faculty of Natural Sciences and Mathematics, University of Montenegro, Podgorica 81000, Montenegro; sladjanak@ucg.ac.me

**Keywords:** heavy metals, proline, chlorophyll, *Trapa natans* L.

## Abstract

The aim of the present study is to investigate the bioaccumulation and translocation of various heavy metals, notably copper, zinc, cadmium and lead, in the different plant organs of *Trapa natans* L. (the root, stem, and leaf) at nine sampling locations, as well as examining the variability in proline accumulation and chlorophyll content due to these heavy metals. Our analysis shows the existence of a statistically significant positive correlation (*r* = 0.75; *p* < 0.05) between the Zn content and the accumulation of proline in the root of the examined species. On the other hand, a statistically significant negative correlation was registered between the content of chlorophyll *a* and the concentration of Zn in the leaf (*r* = −0.68; *p* < 0.05). This indicates that *Trapa natans* L. can be used in biomonitoring Zn-polluted aquatic ecosystems using proline and chlorophyll as sensitive biomarkers.

## 1. Introduction

The pollution of the environment by heavy metals is a result of various industrial activities and is also a multi-element problem in many areas of the world. [1,2,3]. Aquatic plants are known to accumulate various pollutants, including heavy metals, and therefore function as bioindicators of the pollution status of the aquatic ecosystem [4,5].

Certain heavy metals such as copper and zinc are essential micronutrients for plants, because they are involved in numerous metabolic processes as constituents of enzymes and other proteins. However, toxic levels of heavy metals (especially, cadmium or lead) cause several toxic symptoms in plants such as the inhibition of photosynthesis and enzyme activity, growth retardation, disturbed mineral nutrition, water imbalance and the alteration of membrane permeability [6,7].

Increased heavy metal concentrations also cause oxidative stress, including the generation of reactive oxygen species (ROS) [8]. Antioxidant protection is a continuous physiological process in a healthy plant (organism), the objective of which is to prevent the harmful effects of pro-oxidant factors. The antioxidant protection system includes enzymatic and non-enzymatic antioxidants [9,10].

Proline participates in the antioxidant response as one of the non-enzymatic antioxidants. The accumulation of proline in plants is often taken as a biomarker of stress caused by heavy metals [11,12]. Bhupinder et al. (2003) [13] recorded, for the first time, the significant potential of hydrophytes (*Ceratophyllum* sp., *Hydrilla* sp., *Wolffia* sp.) in the accumulation of proline under the influence of different concentrations of Cd^2 +^ ions.

*Trapa natans* L. is a widespread floating aquatic plant, particularly in the northern part of Lake Skadar (Balkan Peninsula). It is characterized by its extremely large biomass, which makes this species potentially important in the phytoremediation process [14,15,16].

Global research on heavy metal removal through phytoremediation using macrophytes is ongoing in a number of countries [17,18,19]. The research activities carried out thus far included the analysis of heavy metal content in the sediments and different macrophytes of Lake Skadar [20,21,22,23,24,25]. However, an analysis of the biochemical characteristics (e.g., proline or chlorophyll) of the different macrophytes in Lake Skadar and their relationship with the content of heavy metals in plants has not been conducted thus far.

In this paper, we have done the research on the very productive area of the Lake Skadar. It is a National park, included in the Ramsar List of international important wetlands; hence, its preservation and protection from pollution is very important. However, intensive industrial and urban development in the region exposed Lake Skadar to anthropogenic pollution by organic and inorganic contaminants, including metals [20].

The aim of the present study was to investigate the bioaccumulation and translocation of the heavy metals Cu, Zn, Cd, and Pb in different parts of *Trapa natans* L. (specifically the root, stem, and leaf) at nine sampling locations, and further examine the variability in proline accumulation and chlorophyll content in the plant due to the presence of these heavy metals.

## 2. Results and Discussion

### 2.1. Distribution of Metals in the Sediment and the Plants

The concentrations of the relevant metals (Cu, Zn, Cd, and Pb) analyzed in the sediments from nine locations around Lake Skadar are presented in Figure 1.

The maximum concentration was recorded for zinc (76.0 mg kg^−1^) at location L4, while the minimum concentration was registered for cadmium (0.03 mg kg^−1^) at location L7. The concentrations of the metals varied in the range of 17–76 mg kg^−1^ for zinc, 4.2–33.1 mg kg^−1^ for copper, 2.73–17.4 mg kg^−1^ for lead, and 0.03–1.18 mg kg^−1^ for cadmium. The values of the metal concentrations in the sediment at all the sampling locations decreased in the following order: Zn > Cu > Pb > Cd.

Similar distributions are noticeable for zinc and copper, with maximum concentrations registered at location L4. The concentrations of cadmium showed the greatest variations and were highest at locations L8 and L9. Lead and cadmium distribution results in the sediment were similar with the highest lead values at locations L9, L3, and L4 and the lowest at locations L2 and L7. The possible anthropogenic sources of metals at locations L8 and L9 could be connected with fish processing, boat traffic, and intensive tourism activities, but the use of fertilizers and chemical products in agriculture may also play a role. At location L1, a relative increase in concentrations was registered for zinc and copper. Anthropogenic sources of the metals at this location are linked to the industrial and municipal waste waters brought by the River Morača, which is the biggest potential polluter of Lake Skadar. However, evaluation of copper and zinc sediment contamination levels by comparison with the existing sediment quality criteria for freshwater sediment, following MacDonald et al. (2000) [26] show that the levels of both copper and zinc are not higher than TEC (the threshold level concentration), indicating that the lake sediments are not currently polluted by copper and zinc deposits.

The distribution of copper, zinc, cadmium, and lead concentrations in the individual part of *Trapa natans* L. are shown in Table 1 for the nine locations.

The maximum concentration was recorded for zinc (42.03 mg kg^−1^) in the root of *Trapa natans* L. at location L1, while the minimum concentration was registered for cadmium (0.03 mg kg^−1^) in the leaf of *Trapa natans* L. at location L2.

The concentrations of copper varied from 1.79 to 5.03 mg kg^−1^, as the highest copper values were found in the root of *Trapa natans* L. at location L9. For lead concentrations were in the range of 0.03–3.68 mg kg^−1^. For the root the highest concentrations of lead in the plant was found at locations L8 and L9, which is similar to cadmium, while the lead concentrations were low at all locations in the stem and the leaf.

The values of the metal concentrations in the individual part of *Trapa natans* L. at all investigated locations decrease in the following order: root> stem > leaf.

The highest concentrations were found in the root of *Trapa natans* L. for all investigated metals (Cu, Zn, Cd, and Pb) and the average concentration in the root was significantly higher (*p* < 0.05) than it was in the leaf or the stem (Figure 2). The finding that the concentration of lead was slightly higher in the leaf than in the stem is maybe unexpected, but it might be a consequence of lead absorption from the atmosphere through the leaf surface [27]. Sawidis et al. (1995) [28] found a very high concentration of zinc (130 mg kg^−1^) in the root of *Trapa natans* L. sampled from Lake Kerkini in Northern Greece. Hoseinizadeh et al. (2011) [29] also found the highest zinc concentration (36.20 mg kg^−1^) in the root of *Trapa natans* L. in Lake Anzali (Iran). Some other aquatic plants have also demonstrated consistently higher metal concentrations in their root than in their stems and leaves [30,31,32].

### 2.2. Translocation Ability (TA)

The mean values of the translocation factor (TA) are presented in Figure 3 and decrease in the following order: Pb> Cd> Cu> Zn.

Most of the examined metals show the greatest translocation from the stem to the leaves. The exception is copper, for which the largest translocation was recorded from the root to the stem. The high values of the translocation factor for lead (TA root/stem = 21.98) indicate the poor mobility of this metal from the root to the aboveground organs. By contrast zinc, cadmium, and copper are extremely mobile, as can be demonstrated by the low values of the translocation factors between root/stem, root/leaf, and stem/leaf: Cd (TA root/stem = 1.90); Zn (TA root/stem = 1.32); and Cu (TA root/stem = 1.49). These results are consistent with the results provided by Mazey and Grem (2009) [31]. For four different macrophyte species (*Najas marina*, *Potamogeton lucens*, *Nuphar lutea*, and *Potamogeton nodosus*) they found the highest mobility in relation to zinc and the lowest for lead (Zn root/stem = 2.46; Zn root/leaf = 1.36; Pb root/stem = 19.8; Pb root/leaf = 19.0).

### 2.3. Proline Content

The values of the proline content in the leaf and the root of *Trapa natans* L. are presented in Figure 4.

Based on the analysis of the obtained results of the proline content in the leaf and root of *T. natans*, it can be seen that the maximum value of proline was recorded in the leaf at locality L1. The lowest proline content was registered in the root at locality L5. The values of the proline in the leaf ranged from 0.68 µmol g^−1^ in the root at locality L5 to 3.1 µmol g^−1^ in the leaf at locality L1. Higher levels of proline content at all localities were recorded in the leaf of the *Trapa natans* L. It is notable that the trend of increasing metal concentrations at certain localities (L1, L4, L6, and L8) is accompanied by a higher content of proline, while the minimum content of proline was registered at locality L5, where we recorded the minimum concentrations of all metals in both the root and the leaf of *Trapa natans* L.

### 2.4. The Relationship between Metal Content in the Plant and Proline Accumulation

A correlation analysis of the metal content (leaf) and the proline accumulation (see Figure 5) showed that there is a statistically significant positive correlation (*r* = 0.75; *p* < 0.05) between zinc content and proline accumulation in the root of the *Trapa natans* L. Many other authors [12,33,34] have also pointed to a significant increase in proline content in different plant species after treatment with different concentrations of heavy metals (for example cadmium, lead, mercury, copper, and zinc).

### 2.5. Chlorophyll Content

The mean concentrations of chlorophyll a, chlorophyll b, and carotenoids are presented in Figure 6. The highest concentration of chlorophyll a was recorded at locality L5 and amounted to 1.15 mg g^−1^, while the minimum concentrations were registered at locations L1, L2, L8, and L9, where the content of chlorophyll a was relatively uniform and ranged from 0.614 to 0.685 mg g^−1^.

By contrast to chlorophyll a, the values of the chlorophyll b concentrations at all the examined localities were of a uniform character, with the maximum recorded at the L3 locality, which was 0.757 mg g^−1^.

### 2.6. The Relationship between Metal Content and Chlorophyll Content in the Plant

The results of our correlation analysis (*p* < 0.05) examining the pigment content and the concentration of the heavy metals in the leaf and root of *Trapa natans* L. are given in Figure 7. This shows that there is a statistically significant highly negative correlation between the content of chlorophyll a and the concentration of zinc in the leaf (*r* = −0.68) of *Trapa natans* L. This type of reduction in photosynthetic pigment under metal stress (from copper, cadmium, and iron) has also been reported by Hou et al. (2007) and Verma et al. (2016) [32,35].

Verma et al. (2016) showed a decreasing trend in total chlorophyll content (of both Chl a and Chl b) from 63.87% to 53.65% in *Trapa natans* L. and *Eichhornia crassipes*, respectively, in iron treatment (40 ppm) after 30 days. However, in cadmium treatment (12 ppm), 56.97% and 55.33% reductions in concentration in the total content were found after 30 days of treatment with respect to the controls in *Trapa natans* L. and *Eichhornia crassipes*, respectively.

The results of this study indicate that *Trapa natans* L. can successfully be used in biological studies in aquatic ecosystems polluted with Zn, by using proline and chlorophyll content as parameters for biomonitoring.

## 3. Materials and Methods

### 3.1. Sample Collection

During the summer of 2012 from nine locations in the Skadar Lake, sediment and p lant materials were taken. Positions and description of the sampling locations are given in Table 2.

The sampling locations (see Figure 8) cover all the major water inflows into the lake (from the Morača River and Crnojevića River) and locations with high potential for local metal contamination. The potential anthropogenic influence of metals into the Lake Skadar ecosystem is from the factories and agricultural areas placed in the Zeta valley. Contaminants from this area are transported to the Lake Skadar by the River Morača (at location L1) and municipal wastewaters of the little towns and villages which surround the northern part of the lake (e.g., locations L8 and L9) as well. From areas of over 25 m^2^ with sufficient abundance of the plant *Trapa natans* L. (see Figure 9), sampling of 10 complete healthy plants was made from each location. The plants with similar size, shape, and weight were taken by hand, packed in bags from polyethylene, and transported to the laboratory. The samples from sediment were collected from same area as plant material. Ekman-type dredge was used for collections of 1kg weight sample of sediment and 0–20 cm layer was taken.

### 3.2. Metal Analysis

The sediment samples were air-dried first and later oven-dried for 48 h at 75 °C. Using the agate mortar the dry samples of sediment were ground and sieved with a 1.5 mm sieve. Mineralization of around 0.5 g sediment sample was performed using microwave digestion with a mixture of HCl: HNO_3_ (3:1). With 2M HNO_3_, the solutions were diluted to a planned volume of 100 cm^3^. The sample material was washed using deionized water in the laboratory to remove periphyton and detritus. The plants samples were divided into the three groups roots, the stems, and the leaves. Then we dried samples for 48 h at 75 °C. Electrical mill grinding of dry samples was performed into a fine and homogenized powder. Mineralization of the prepared samples 0.5 g were done with a Milestone Microwave Ethos 1, and mixture of HNO_3_ and H_2_O_2_ (3:1). When digestion is completed, using deionized water the solution was diluted to a planned volume of 50 cm^3^. Determination of the metals (Cu, Zn, Cd, and Pb) concentrations in the plant and sediment samples was done using inductively coupled plasma optic emission spectroscopy (ICP-OES) on a Spectro Acros instrument. Following the Sigma Aldrich solutions, the measurements standards were applied. Evaluation of the analytical method reliability was done by the analysis of the certified standard reference materials NCS DC73348 (Bush Branches and Leaves) and NCS DC70312 (Tibet sediment) from the China National Analysis Center for Iron and Steel in Beijing. All the sediment and plant samples were prepared in triplicate and their mean values were considered. Results are presented based on dry weight.

### 3.3. Calculation of Translocation Ability

The translocation ability (TA) is used for estimation of metals transfer within the plant (from stem to leaf and from root to stem). The translocation ability (TA) was calculated as the ratio of concentration of metal between the individual parts of each plant, from the lower to the upper part of the plant (TA = Metal root or stem/Metal stem or leaf). A lower TA means a higher translocation ability [36].

### 3.4. Determination of Proline

The proline content in the plant biomass sample was estimated following the procedure described by Bates et al. (1973) [37]. About 0.5 g of the fresh plant sample was homogenized in 10 mL of 5% sulfosalicylic acid and centrifuged at 12,000 rpm for 10 min. In a test tube, 2 mL of filtrate, 2 mL of glacial acetic acid, and 2 mL of ninhydrin reagent were added. We covered the test tube with aluminum foil and heated it in a water bath at 100 °C for one hour. A brick red color appeared. Then 4 mL of toluene was added and then transferred to a separating funnel. After thorough mixing, we took the absorbance at 520 nm using the toluene as a blank. The proline content is expressed in μmol (proline)/g of fresh material.

### 3.5. Photosynthetic Pigment Determination in Leaves

About 0.5 g of fresh plant sample was homogenized in 10 mL of pure acetone with the addition of quartz sand. To prevent the acidification of the sample due to the acid secreted during maceration and to ensure the stability of the chlorophyll molecule, the mortar may also contain a small amount of calcium bicarbonate. After homogenization, the extract was transferred to a glass filter and the water jet on the pump was switched on to filter the extract into a test tube contained in a vacuum flask. The total volume of the extract was 25 mL (the extract was poured from a test tube into a measuring vessel, made up with acetone to the correct volume, and then shaken).

After extraction in pure acetone, the pigment extract was diluted as needed, and the light absorptions were measured on a spectrophotometer at appropriate wavelengths (662 nm, 644 nm and 440 nm). The pigment concentration in mg/L was calculated according to Wettstein (1957) [38] after which it was converted to mg/g of fresh material.

### 3.6. Statistical Analysis

The results obtained were analyzed using the statistical software package Statistica for Windows 7.1. (StatSoft Inc., 2006, Chicago, IL, USA). A non-parametric ANOVA with a significance level of *p* < 0.05 was carried out for the content of each metal in the root, the stem, and the leaf. The Kruskal–Wallis nonparametric test (*p* < 0.05) and the post hoc Tukey test (*p* < 0.05) were used for the testing of statistically significant differences between groups. The correlation between the content of the metal in the specific plant parts (the root, the stem, and the leaf), the proline accumulation, and the chlorophyll content were assessed by the Pearson’s correlation coefficient (*p* < 0.05).

## 4. Conclusions

Various studies of the metal contents (for copper, zinc, cadmium, and lead) in the vegetative organs of *Trapa natans* L. showed that this species has great potential in terms of metal accumulation and can be of great importance in the biomonitoring of aquatic ecosystems. One very significant point recorded in our examination is that the maximum concentrations for all the investigated metals were registered in the root of the *Trapa natans* L., which may be an indication that this species possesses certain specific mechanisms of metal binding at the root level. Moreover, a statistically significant positive correlation (*r* = 0.75; *p* < 0.05) between the zinc content (as the most abundant metal) and the accumulation of proline in the root of the *Trapa natans* L. suggests that there is a possibility that in the case of *Trapa natans* L., the proline may serve as a biochemical indicator of the degree of contamination of the aquatic environment exposed to the direct influence of heavy metals. On the other hand, a statistically significant negative correlation was registered between the content of chlorophyll a and the concentration of zinc in the leaf (*r* = −0.68; *p* < 0.05). This indicates that *Trapa natans* L. can be used in biomonitoring Zn-polluted aquatic ecosystems using proline and chlorophyll as sensitive biomarkers.

## Figures and Tables

**Figure 1 plants-09-01287-f001:**
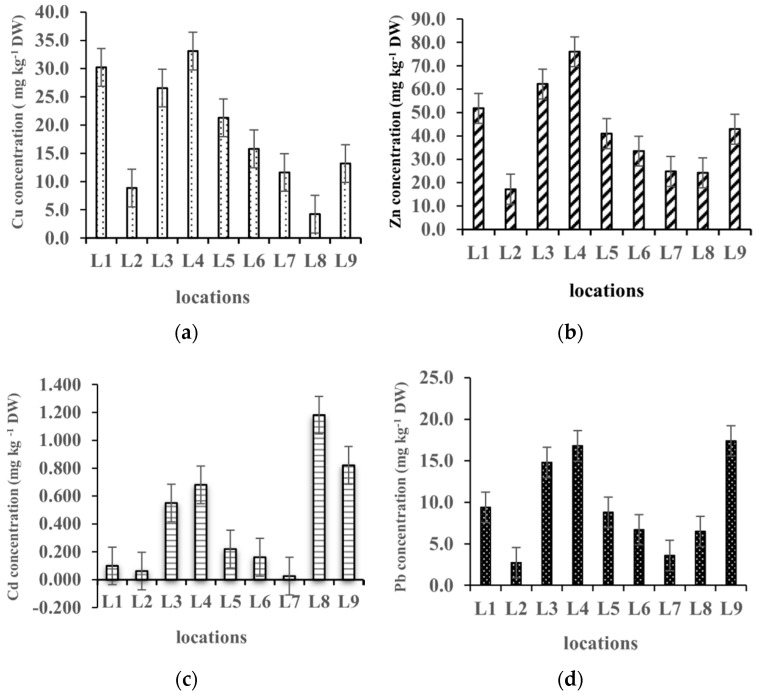
Concentrations of Cu (**a**), Zn (**b**), Cd (**c**), and Pb (**d**) in the sediment samples from different locations.

**Figure 2 plants-09-01287-f002:**
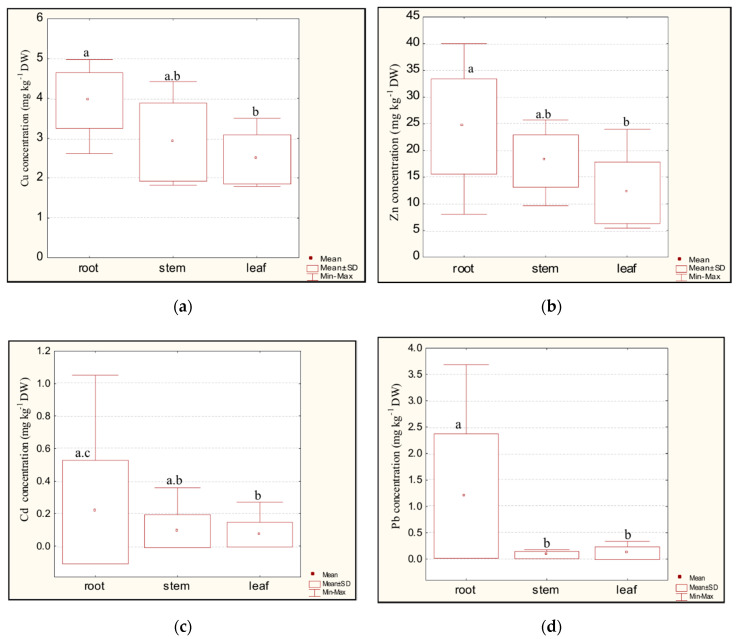
Box-plot graphs of Cu (**a**), Zn (**b**), Cd (**c**), and Pb (**d**) concentrations in the individual parts of *Trapa natans* L. at nine sampling locations (box-plot boundaries indicate the average value, standard deviations, and minimum and maximum value). The statistically significant differences among groups according to the post hoc Tukey’s test (*p* < 0.05) are indicated by the different letters.

**Figure 3 plants-09-01287-f003:**
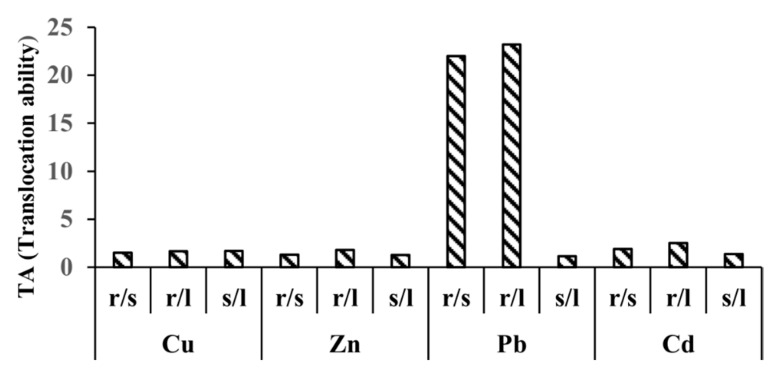
Ranslocation ability(TA_root/stem; root/leaf; stem/leaf_) for Cu, Zn, Cd and Pb.

**Figure 4 plants-09-01287-f004:**
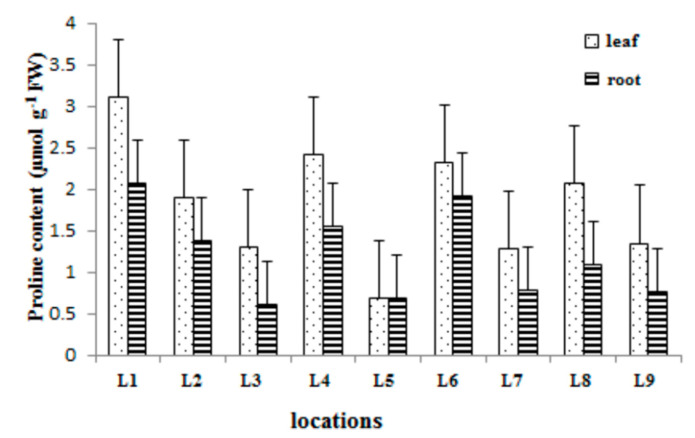
Proline accumulation in the leaf and the root of *Trapa natans* L. sampled from different locations.

**Figure 5 plants-09-01287-f005:**
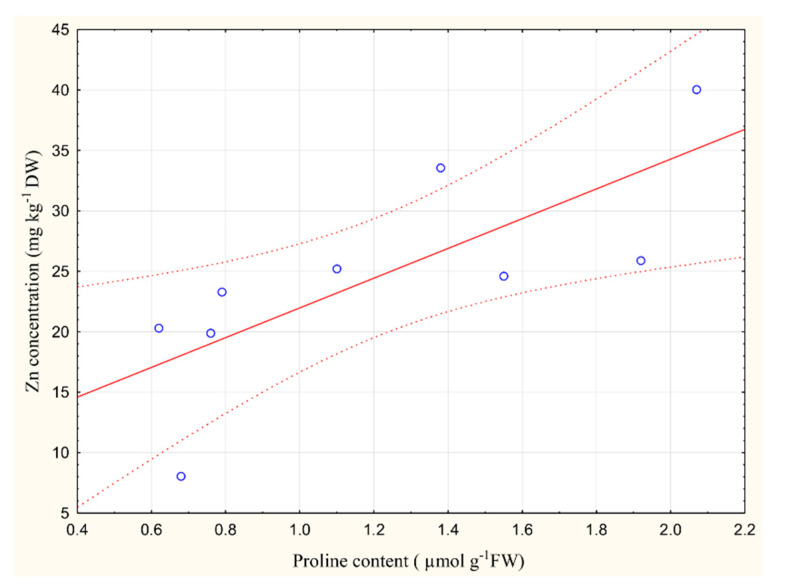
The values of the Pearson’s correlation coefficient (*p* < 0.05) between Zn content and proline accumulation in the root of *Trapa natans* L.

**Figure 6 plants-09-01287-f006:**
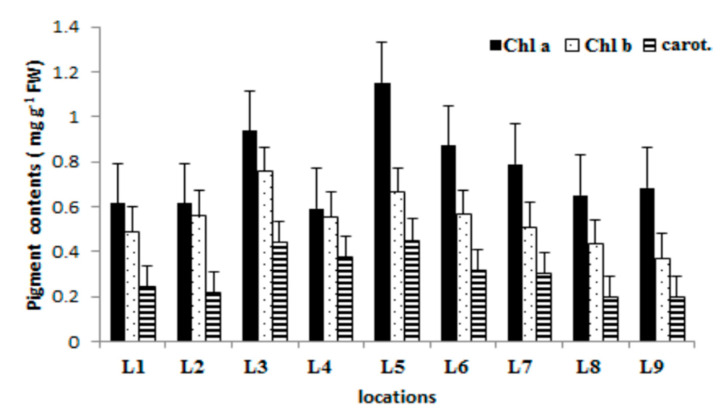
Content of chlorophyll (chl a, chl b) and carotenoid (carot.) in *Trapa natans* L. sampled from different locations.

**Figure 7 plants-09-01287-f007:**
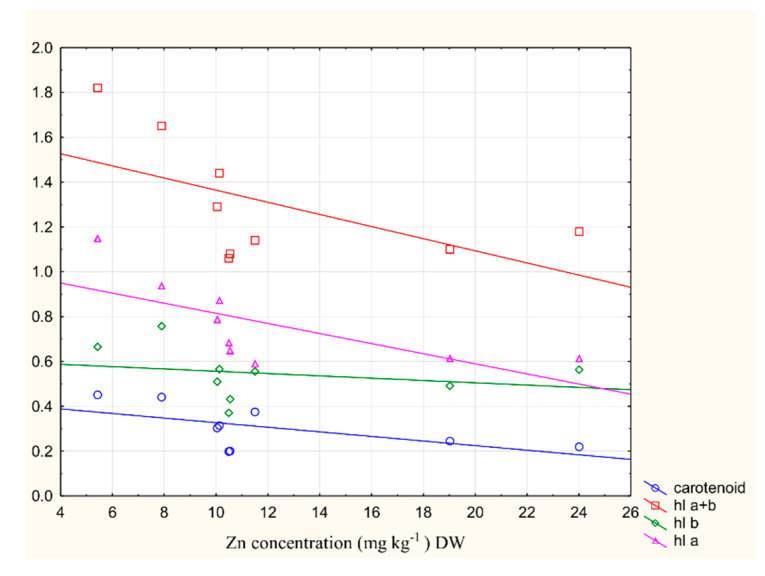
The values of the Pearson’s correlation coefficient (*p* < 0.05) between the pigment content and the Zn concentration in the leaf of *Trapa natans* L. *r* = −0.68, *p* < 0.05.

**Figure 8 plants-09-01287-f008:**
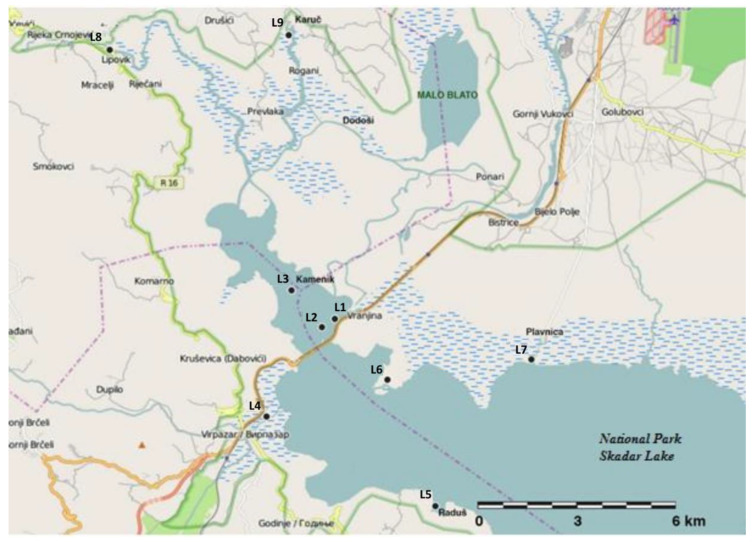
Map of Lake Skadar with the sampling locations.

**Figure 9 plants-09-01287-f009:**
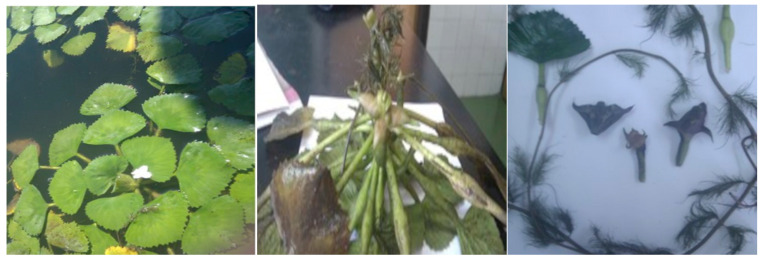
Illustrative photos of typical rosettes, leaves, and stem of *Trapa natans* L.

**Table 1 plants-09-01287-t001:** The concentration of heavy metals (mg kg^−1^) in the individual parts (root, stem, leaf) of *Trapa natans* L.

Metal	Min–Max (mg kg^−1^)		Average ± SD	CV (%)
**Cu**	root	2.62–5.03	3.949 ± 0.700 ^a^	17.73
	stem	1.82–4.66	2.906 ± 0.980 ^ab^	33.72
	leaf	1.79–3.67	2.473 ± 0.620 ^b^	25.07
**Zn**	root	8.03–42.03	24.53 ± 8.925 ^a^	36.38
	stem	9.66–26.02	18.04 ± 4.906 ^ab^	27.20
	leaf	5.43–24.31	12.11 ± 5.759 ^b^	47.56
**Cd**	root	0.066–1.12	0.210 ± 0.320 ^ac^	152.38
	stem	0.042–0.373	0.092 ± 0.101 ^ab^	109.78
	leaf	0.032–0.287	0.073 ± 0.075 ^b^	102.74
**Pb**	root	0.030–3.684	0.113 ± 0.123 ^a^	108.85
	stem	0.060–0.396	0.079 ± 0.069 ^b^	87.34
	leaf	0.060–0.396	1.194 ± 1.177 ^b^	98.58

The different letters denote significant differences between the particular metal concentrations in the individual parts of plant (*p* < 0.05).

**Table 2 plants-09-01287-t002:** Description and position of the sampling sites.

Location	Description	Latitude and Longitude
L1	Inflow at the right branch of the River Morača	42°27′70″ N; 19°12′32″ E
L2	Small lake at the right branch of the River Morača	42°27′57″ N; 19°12′15″ E
L3	Kamenik	42°28′54″ N; 19°10′ 25″ E
L4	Milovića Bay	42°26′ 22″ N; 19°10′77″ E
L5	Underwater spring at Raduš,	42°22′79″ N; 19°12′94″ E
L6	Inflow at the left branch of the River Morača	42°25′84″ N; 19°13′46″ E
L7	Inflow of the River Plavnica	42°26′55″ N; 19°19′80″ E
L8	Crnojevića River	42°35′23″ N; 19°03′99″ E
L9	Karuč	42°35′81″ N; 19°10′71″ E

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
