# Peer review of "The Effect of Cu, Zn, Cd, and Pb Accumulation on Biochemical Parameters (Proline, Chlorophyll) in the Water Caltrop (Trapa natans L.), Lake Skadar, Montenegro"

_plants, 2020, doi:10.3390/plants9101287_

Round 1
Reviewer 1 Report
The work examines of the bioaccumulation and translocation of the heavy metals Cu, Zn, Cd and Pb in different parts of Trapa natans L. (specifically the root, stem and leaf) in a protected area, and further examine the variability in proline accumulation and chlorophyll content in the plant due to the presence of these heavy metals. The work can be considered original and scientifically significant.
Comments:
Edit the structure of the work: 1 Introduction, 2 Material and Methods, 3 Results and Discussion, 4 Conclusions
Table 1 Add Notes: CV
Fig. 2 Add Notes: a, b, c
edit units:µmol g-1, mg kg-1, mg g-1
edit formulas: HNO3, H2O2
Author Response
Dear reviewer,
In attachment is word document with answers to your comments.
best regards
Dragana Petrovic

Reviewer 2 Report
This paper present some interesting results regarding the effects of copper, zinc, cadmium and lead in a plant species (Trapa natans). The paper is generally well written and the results are properly presented.
Some general formatting problems were detected in the paper and should be corrected (eg, paragraph starting in line 36, final period is missing or is misplaced). Check all the subscripts and superscripts (include in chemical formulae). There should be a space between numbers and units (check all the text).
Title: I would replace “heavy metals” in the title with “copper, zinc, cadmium and lead”
Figure 1 – the “-1” subscript in kg is not correct
In the figure the amount of Cd is not visible, the authors should use a different figure for Cd due to the difference in scale.
Line 65 – Please place the “-1” as a superscript (check all the text, this error appears elsewhere).
Table 1 – There is an extra “z” in front of Zinc. Also please use the same size of letter for the a and b superscripts.
Line 116 – title should be “Translocation ability (TA)”
It would be better to present the TA values in a small table or figure instead of the text.
Line 129 – should be “proline”
Section 2.4 – A justification should be given for the observed correlation between Zn and proline (and the same should be done for the correlations between metals and chlorophyll).
Line 230: The title should be “Determination of proline”
Author Response

(The authors gave the same response as above.)

Reviewer 3 Report
The manuscript describes the potential use of Trapa natans L. as bioindicator of heavy metal pollution in the areas of Skadar Lake in Montenegro. The topic is interesting since heavy metal contamination is a big problem in many areas of the world for ecosystems, having a big impact also on human health.
Despite the importance of the topic, I think that the manuscript needs to be reorganized and improved and cannot be accepted as submitted.
My major observations below:
Abstract:
- I recommend using verbs in the past, instead of the present
- content and concentration are not the same: please check the correctness in all the text.
- line 12: I suggest writing "in the different plant organs of Trapa natans L. at nine sampling locations..."
- line 18: if the chlorophyll a concentration is low when the zinc accumulation is high, I think that this is an objective observation derived from the analyzed data, therefore is not a hypothesis as wrote the Authors “is the possibility”.
Introduction:
In my opinion the introduction should be improved with additional information on the studied heavy metals and macrophytes to better explain the importance of this work and enhance it.
- line 24: I suggest adding “of the world” after areas
- lines 27-30: I agree that toxic levels of Cd and Pb negatively affect plant growth and physiology, - but also Cu and Zn are heavy metals and an excess of them can cause severe damage to plants, although they are required at low level since they are essential micronutrients for plant. On the basis of these observations, I suggest rewriting this part.
- line 38: delete “.” after reference n 13 and check the spacing
- line 40: Cd2+ instead of Cd2 + (please correct this kind of error in all the text, check the chemical formula)
- line 42: I suggest adding in brackets “Balkan Peninsula” after “Skadar”
- line 55: “was” instead of “is”
Results and discussion:
- I recommend enlarging the Figure 1 so that the histograms are clearly visible; moreover, I suggest adding different letters or asterisks to indicate significant differences (for metal concentrations) between the various sampling locations. In my opinion the map reporting the 9 sampling locations could be moved in the supplemental information (if possible)
- line 67: please check the correctness of “mgkg-1” in this paragraph and add a space between the unit and “for”
Par 2.2: I cannot find figure or table reporting data of translocation ability
Discussion should be implemented.
Materials and Methods:
Par 3.2: Please specify how many samples for each sediment were used for metal concentration determination. Moreover, specify how many plants were analyzed per each location. Did these plants also used for the analyses of all the other parameters (proline, chlorophyll concentration)?
Par. 3.5: I suggest changing the title with “Photosynthetic pigment determination in leaves”, since not only chlorophylls but also carotenoids were recorded. I recommend the authors also to consider the method and the equations of Porra 2002 for the chlorophyll determination. How long did the leaf tissue remain in the extraction solvent? In general, this step should occur in the dark at 4 °C.
References:
The genus and species should be written in italics: please check the correctness.
Journal title should be formatted in accordance to the journal standard: all in full or all abbreviated.
Author Response

(The authors gave the same response as above.)
